# Personal Narratives in Trauma-Related Disorders: Contributions from a Metacognitive Approach and Treatment Considerations

**DOI:** 10.3390/bs15020150

**Published:** 2025-01-30

**Authors:** Courtney N. Wiesepape, Elizabeth A. Smith, Andrew J. Muth, Laura A. Faith

**Affiliations:** 1Austin VA Clinic, Veterans Affairs Central Texas Health Care, Austin, TX 78744, USA; 2Department of Psychology, Indiana State University, Terre Haute, IN 47809, USA; liz.smith@indstate.edu; 3Portland VA Medical Center, Portland, OR 97239, USA; andrew.muth@va.gov; 4Richard L. Roudebush VA Medical Center, Indianapolis, IN 46202, USA; laura.faith@va.gov

**Keywords:** trauma, post-traumatic stress disorder, PTSD, personal narrative, narrative identity, metacognition, psychotherapy, treatment, MERIT

## Abstract

Traumatic experiences are significant life events that are thought to impact one’s personal life narrative and narrative identity. Individuals who have experienced trauma may display fragmented memories and decreased narrative cohesion, resulting in trauma narratives that are disjointed and poorly integrated into the larger picture of their lives. Metacognition, defined as the ability to form increasingly complex and integrated ideas about the self, others, and the wider world, offers a framework to better understand life narratives and has been proposed as a construct that allows personal narratives to evolve in response to new experiences. In this paper, we will review the alterations commonly observed in trauma narratives. We will utilize the integrated model of metacognition as a framework to understand these deviations with an eye toward clinical implications. Although treatments that focus on trauma narratives exist, we argue that the study of metacognition provides unique insights into the process of integration of traumatic experience into an evolving personal narrative and may allow for more complete treatment of trauma-related disorders.

## 1. Introduction

Post-traumatic stress disorder (PTSD) falls under the Trauma- and Stressor-Related Disorders section of the Diagnostic and Statistical Manual of Mental Disorders Fifth Edition, Text Revision (DSM-5-TR). To be diagnosed with PTSD, an individual is required to have been exposed to a traumatic event (i.e., death, threatened death, actual or threatened serious injury, actual or threatened sexual assault) through direct exposure, witnessed the trauma, learned that the trauma happened to a close other, or been indirectly exposed to details of trauma (e.g., first responders). In addition, there are four symptom clusters that must be present in varying levels, including (1) re-experiencing (e.g., nightmares, flashbacks), (2) avoidance of trauma-related stimuli, (3) negative thoughts or feelings (e.g., negative assumptions about oneself and the world, negative affect, decreased interest in activities), and (4) trauma-related arousal and reactivity (e.g., irritability, aggression, hypervigilance). Symptoms must have lasted at least one month and cause significant distress or functional impairment ([3]). Two specifiers are included for individuals who experience delayed onset of symptoms (>6 months after the event) and individuals with dissociative symptoms, including, in particular, depersonalization and derealization.

The International Statistical Classification of Diseases and Related Health Problems (11th ed.; ICD-11; [71]) diagnostic criteria for PTSD include three symptom clusters similar to those listed in the DSM-5-TR: re-experiencing the traumatic event, avoidance of reminders, and hyperarousal symptoms that indicate a persistent sense of threat. To meet criteria for the diagnosis, one symptom from each cluster must be present for several weeks after trauma exposure and cause functional impairment.

Although most individuals are thought to have experienced a traumatic event in their lifetime, only a percentage (<10%) develop PTSD ([8]). For example, in the international WHO World Mental Health surveys, 70.4% of individuals experienced lifetime trauma, but the lifetime prevalence of PTSD is estimated to be 6.8% ([31]). However, recent research suggests that trauma-related disorders are becoming progressively more common with increases in the prevalence of both PTSD and acute stress disorder (ASD) in US college students ([73]). The average duration of any PTSD symptom is 72.3 months; however, up to 40% of individuals with PTSD fully recover (i.e., experience no clinically significant symptoms) within one year ([31]). One meta-analysis of 78 studies found that recovery in PTSD primarily occurs in the first three months after the experience of trauma ([16]).

Another meta-analysis found that even without specific, PTSD-focused treatment, an average of 44% of individuals with PTSD were in remission at follow up (mean follow up at 40 months), with the highest rate of remission found in studies on PTSD in individuals who experienced natural disasters ([46]). Higher rates of PTSD have been found after exposure to interpersonal trauma (e.g., sexual assault, intimate partner violence) compared to non-interpersonal trauma (e.g., natural disasters), with trauma involving interpersonal violence having the highest associated risk ([31]). Taken together, these findings suggest that non-interpersonal trauma has higher rates of remission and lower rates of PTSD compared with interpersonal trauma. It is likely that interpersonal trauma affects people differently than non-interpersonal trauma, and treatments must reflect this difference. For instance, self-blame ([33]) and trauma associated with a high level of betrayal ([41]) may be important in the etiology and maintenance of interpersonal trauma.

For individuals who do not recover without intervention, various treatment options exist. Results of efficacy studies focused on interventions to prevent the development of PTSD are mixed at best, with some prevention treatments (e.g., debriefing) even found to be harmful ([29]). Although research is limited, one meta-analysis concluded that early intervention may lead to lasting effects in the improvement of PTSD severity but cited limited data as a barrier to more definite conclusions ([51]). In one early intervention study, both cognitive therapy and prolonged exposure were found to be effective in decreasing PTSD rates at a 5-month follow up when administered approximately one month after a traumatic event. Delayed prolonged exposure (~12 weeks after the traumatic event) was also found to be effective; however, at a 9-month follow up, there were no differences between the prolonged exposure and waitlist groups ([60]), further indicating that a portion of individuals diagnosed with PTSD go on to recover without intervention. Meta-analyses have consistently found that cognitive therapy, exposure therapy, and eye movement desensitization and reprocessing (EMDR) treatments are effective in treating PTSD ([69]). A more recent meta-analysis specifically found efficacy of exposure therapies (e.g., prolonged exposure, narrative exposure therapy), cognitive therapy (CT) and cognitive processing therapy (CPT), and EMDR ([13]).

Despite promising recovery outcomes both with and without treatment, a portion of individuals diagnosed with a trauma-related disorder experience a more chronic course. In fact, it is estimated that approximately 39% of individuals diagnosed with PTSD experience a chronic course ([57]) and may experience symptoms of PTSD even after 10 years ([32]). Typically, PTSD symptomology lasting one to three months is classified as acute, whereas PTSD symptomology lasting more than three months is considered to be a chronic course ([11]). Chronic PTSD may be due to a number of factors, including unremitting symptoms, lack of natural remission, or delay in treatment. For instance, one study found that delay in PTSD treatment was 2.5 years for post-911 veterans, 16 years for pre-911 veterans, and 15 years for civilians ([20]). In addition to chronic courses of PTSD, there is some evidence that complex PTSD (cPTSD) is more common and more debilitating than PTSD ([28]). Although cPTSD shares features with other psychiatric conditions (i.e., PTSD, borderline personality disorder), it is thought to be a disorder with a distinct presentation and treatment considerations ([14]; [19]). The diagnosis of complex PTSD may include individuals who have experienced repetitive trauma from which escape is difficult or impossible and more severe symptoms in the following domains: behavioral, emotional, cognitive, interpersonal, and somatization ([14]). Relatedly, cPTSD may include dysregulation in emotion processing, self-organization, and relational security in addition to core PTSD symptoms ([19]). The disorder was formalized as a diagnosis in ICD-11 and requires the core symptoms of PTSD as well as disturbances in emotion regulation, self-concept, and relationships as criteria. Furthermore, ICD-11 notes that additional clinical features may include suicidality, substance abuse, depressive symptoms, psychotic symptoms, somatic complaints, and an inability to maintain some level of functioning without significant additional effort ([71]). Thus, treatment of cPTSD may be more complex or long-term when compared to traditional PTSD treatment.

The question of why some, but not all, who experience trauma develop a trauma-related disorder has served as an important catalyst for the development of models of PTSD and treatment approaches. [17] ([17]) addressed this question in their seminal cognitive model of PTSD, which holds that all traumatic events produce PTSD symptoms in the short term and that most experience a natural remission of these symptoms with time. This natural recovery process involves making sense of the event in terms of reasons for the event happening, evaluating one’s response to the event, and identifying implications of the event for one’s broader life. Within this model, a subset of traumatized individuals will experience persistent symptoms, partly as a result of cognitive factors that serve as barriers to natural recovery. Specifically, some individuals develop highly idiosyncratic appraisals of the event and their response to the event that function as a means to prevent similar experiences in the future. However, these appraisals are often self-blaming, involving excessive focus on what the person “should” have done differently in order to avoid the event, and assume greater levels of control over external events than is accurate.

Certain factors present challenges for the effective treatment of PTSD. [18] ([18]) summarized the existing data on the efficacy of cognitive therapy for PTSD, and although the findings were largely favorable, they identified barriers that may contribute to less favorable outcomes, including clients’ experiences of social problems, which may take session time away from discussing discrete factors related to the trauma, as well as re-experiencing symptoms. Additionally, they noted that treatment may be more complicated to deliver among patients with multiple events in their trauma history. Clients with multiple trauma exposures are subject to maladaptive (e.g., self-blaming) attributions becoming reified with each subsequent traumatic experience (i.e., “I attract bad people”). This may lead to particularly extensive restriction of activities, such as socializing with others, resulting in profound functional impairments and social withdrawal. In addition, other clinical factors may complicate the treatment of PTSD, including exposure to childhood adverse events, personality pathology, female gender, and combat exposure, which may also be related to traumatic brain injury. Overall, these results suggest that although existing trauma treatments are effective for many, those whose traumas impacted their view of themselves outside of a single event (e.g., multiple traumas) or their position in the larger social world may need additional approaches that address these more integrative and complex issues.

For some individuals who have experienced pervasive trauma, stress, or adversity, the boundaries between the “traumatic event” and the larger flow of life may be difficult to draw, which complicates knowing how to target interventions. While failing to integrate a traumatic event into one’s larger life story implies more difficulty involved in coping with the trauma, making traumatic events a central part of a life story and relating events to one’s sense of self could have worse implications for long-term recovery ([56]). Thus, it is important for people who experience trauma to navigate how to integrate a traumatic event (or multiple traumatic events) into their life story in such a way that they can make sense of the event, integrate it into their sense of self and position in the world, and move forward with their life. One way to conceptualize this integration of an event into the larger sense of oneself is by exploring narrative identity. In this paper, we examine the impacts of trauma on personal narratives and explore the role of metacognition both as a lens through which to understand this connection and as a potential avenue for treatment.

## 2. Methods

To explore the intersection of trauma, personal narratives, and metacognition, we conducted a theory-driven narrative review. In this review, we aim to introduce the construct of narrative identity and review key findings related to alterations in narratives found in individuals who have been exposed to trauma. We next seek to introduce the integrated model of metacognition and explore its proposed connections to narrative identity with special attention paid to metacognitive deficits that have been observed in trauma-related disorders. After reviewing support for (1) changes in narratives in individuals exposed to trauma, (2) the connection between metacognition and narrative identity, and (3) the existence of metacognitive deficits in trauma-related disorders, we offer an exploration of treatment implications with a focus on narrative treatment. Specifically, we review an existing narrative-oriented metacognitive intervention that has been shown to aid in recovery, improve metacognition, and impact narrative identity. In summary, the primary aim of this paper is to explore the intersection of narrative identity and metacognition in the context of trauma-related disorders with an eye toward treatment implications. This paper is intended for an audience of clinicians and researchers alike to offer both practical information for trauma interventions and to act as fodder for future empirical studies.

Relevant articles were identified through searches of electronic databases using keywords, such as trauma, post-traumatic stress disorder, narrative, narrative identity, metacognition, and treatment. Additional articles were identified through manual review of reference lists in relevant publications. A broad range of articles were included to fully explore this topic, including case studies, reviews, theoretical work, and empirical studies. Included studies displayed strong relevance to the topic and used qualitative, quantitative, or mixed methods. Other inclusion criteria included articles being peer-reviewed and published in English. Studies were excluded if they did not focus on a primary topic relevant to our review, if they were not peer-reviewed (e.g., dissertations, conference abstracts), or if they lacked methodological transparency to assess relevance to our aims. The authors had expertise in the areas of metacognition and trauma-related disorders and drew on existing knowledge of the integrated model of metacognition to guide the review and ensure accuracy in its application to trauma-related disorders. This approach ensured that the review captured diverse perspectives and insights into the interplay between trauma, personal narratives, and metacognition.

## 3. Narrative Identity

[42] ([42]) first introduced a framework for studying the self through narrative identity. In this framework, identity is constructed through personal narratives or life stories, which are shaped by personal experiences and broader cultural norms. Numerous aspects of personal narratives can be studied, including their structure and content, their function, and their relationship to psychosocial health. These personal narratives tell the story of one’s life, including rich histories and how one came to be the person they are today. Individuals often balance themes of agency (achievement, power) and communion (relationships, connection) in personal narratives ([43]). In addition to these themes that allow individuals to interpret and organize life experiences, personal narratives provide a sense of continuity in the flow of life from one’s past, one’s purpose in the present moment, and a direction for the future ([72]). Essentially, the development of personal narratives relies on the ability to interpret and make sense of events as they unfold, organizing them within a temporal framework that is structured through the pursuit of specific goals ([62]). Finally, recalling past autobiographical memories and determining how they relate to the larger context of an evolving narrative is an important aspect of developing a life story ([44]). More recent research has found a reliable three-factor structure of narrative identity, which includes (1) motivational and affective themes, (2) autobiographical reasoning, and (3) structural aspects ([45]). In this model, motivational and affective themes were most related to well-being.

Narrative reconstruction is often thought of as a key task in recovery from PTSD and other trauma-related disorders. One study utilizing thematic analysis of interviews with mental health peer support workers found that self-mastery and personal agency are particularly important aspects of narrative identity reconstruction ([30]). The reconstruction, or re-storying, of personal narratives in the aftermath of trauma has also been connected to post-traumatic growth ([49]; [25]), meaning making, and the identification of purpose ([27]).

### Narrative Identity in Trauma-Related Disorders

Alterations in narrative identity typically occur within specific domains, which are most commonly narrative structure and narrative themes. Although less often explored, changes in narrative responsivity, or how a narrative changes in response to new events, are also vital to understanding how changes to personal narratives impact narrative identity and recovery. See Table 1 for a summary of alterations in narratives in trauma-related disorders.

Some research suggests that trauma memories are fragmented and incomplete immediately after a traumatic event and generally increase in coherence with time. In one study that observed changes in the structure of trauma narratives over time, narrative structure was measured soon after a traumatic experience during an emergency department visit and at a one-year follow up. This study showed improved narrative structure over time without intervention and that the baseline narrative structure was associated with fewer depressive and posttraumatic symptoms at follow up. Specifically, interpretative elaboration and coherence of narratives at baseline predicted positive changes in coping strategies ([6]). Another study followed five individuals admitted to the emergency room after surviving a shooting ([64]). Each shared their account of the event at multiple time points. Immediately after the event, memories were incomplete and fragmented and included intensive, sensory details. Accounts of the event were given again after one week, one month, and four months and demonstrated a general trend toward increased coherence over time.

Narrative coherence is an important aspect of narrative structure and is often conceptualized as the degree to which a narrative “makes sense” and conveys the content and meaning of events in a structurally cohesive manner ([66]). Coherence can be measured in a variety of ways, including using linguistic measures (e.g., semantic associations) and coding systems (e.g., Narrative Coherence Coding Scheme). These coding systems often include timing, chronologicity, number indicator words (e.g., the next day), casual links, etc. ([53]). Overall, narratives of traumatic experiences tend to be less coherent than narratives of other key life events or memories ([66]). In addition, neutral narratives tend to be more coherent (i.e., increased semantic association between words) than trauma narratives ([68]). However, in one study, thematic coherence was found to be related to higher anxiety levels, with the authors suggesting that narrative coherence may reflect incomplete attempts at meaning making. [56] ([56]) discussed the integration of a traumatic memory into one’s overall life narrative and argued that overly integrated trauma narratives are associated with PTSD more so than lack of integration. That is, the more a person experiences the event as central to their life story and self-concept, the greater the risk for mental health symptoms ([55]). This suggests that while integration is important, if the trauma becomes central to one’s narrative identity, it can be detrimental to recovery.

While fragmentation in the acute stages following trauma is common, evidence for longstanding fragmentation of trauma memories is mixed ([54]). A review of trauma narratives in PTSD found that disrupted temporal context and sensory dominance in language were related to trauma-related pathology; however, the connection between trauma-related symptoms and narrative fragmentation was inconclusive ([50]). A review of 22 studies of trauma narratives reported mixed findings and suggested that fragmentation may occur at similar levels compared with other mental disorders ([12]). Another study compared levels of fragmentation in traumatic memories in the year after the event with memories of a pleasant event and an important event; the traumatic memories were no more fragmented than memories of the other events ([54]).

Other studies have suggested that narrative content is more strongly associated with trauma-related reactions. Namely, more emotion words, higher cognitive process, and less self-focus were found to be related to lower symptomology ([24]). A small number of studies have examined variations in meaning making as predictors of PTSD symptoms. The research suggests that while meaning making reflects a natural, common part of trauma recovery, the extent to which people frame meaning in terms of a personal, subjective response is protective against developing PTSD symptoms ([64]). Qualitative studies are especially relevant to observing changes in personal narratives. One study used grounded theory analysis to explore interviews of trauma survivors enrolled in psychotherapy with focus on the process of trauma recovery. The study found two major themes, including increased trauma narrative coherence and a more reflective stance toward one’s personal history related to trauma recovery ([26]).

These studies highlight the importance of meaning-making processes to the overall course of recovery after trauma. Specifically, as people recover after trauma, they make attempts to make sense of what happened and what it means to them. Individuals who find difficulty in identifying nuanced, personally relevant meanings and integrating traumatic events into one’s life narrative in a balanced manner may be at increased risk for developing PTSD symptoms. PTSD treatments that include components of meaning making may be especially helpful for those with chronic PTSD who have difficulty integrating trauma experiences into their life and narrative identity.

## 4. Metacognition and Narrative Identity

Recently, the construct of metacognition has been proposed as a key aspect of narrative identity and a mechanism for changes in life narratives. [38] ([38]) proposed that intact metacognition is not only important for recovery but also for the evolution of narratives for individuals diagnosed with psychosis and related disorders. From this perspective, deficits in metacognitive functioning interfere with the integration of new experiences into the larger life narrative and one’s understanding of the self, others, and the wider world. This then compromises an individual’s ability to make sense of how various experiences may be related to one another. For example, a new experience that contradicts one’s previous understanding of related events, themselves, or the world (e.g., trauma) may become incomprehensible or confusing and unable to be integrated into the ongoing flow of life. This may result in an understanding of the self, others, and the wider world within one’s life narrative that does not evolve or is impervious to change short of dramatic experiences.

Broadly, cognition includes a wide range of mental processes and abilities used to gain knowledge, including problem solving, perception, decision making, and planning ([4]). Metacognition falls under the umbrella of cognition and refers to the capacity to think about one’s thinking. The integrated model of metacognition has expanded upon this to include the awareness of and ability to reflect upon thoughts, emotions, and behaviors of the self and others ([47]). Importantly, there are four foci of metacognition: self-reflectivity, understanding the mind of the other, decentration, and mastery. Self-reflectivity is the ability to understand the self in increasingly complex and integrated ways. This may range from the ability to identify a discrete thought or emotion to narrating a complex life story that includes many integrated internal experiences and various linked themes related to understanding oneself. Understanding the mind of the other is similar to self-reflectivity but with a focus on the other. Thus, this domain of metacognition involves understanding specific others (e.g., mother, best friend) in increasingly complex and integrated ways. Decentration is defined as the ability to understand that one’s community is a collective of individuals with unique histories, challenges, and experiences that are typically unrelated to the self. It ranges from the ability to understand that others have lives outside of one’s own to understanding the influence of broader cultures. Finally, mastery is the ability to use metacognitive knowledge about the self, others, and the broader community to understand and manage psychosocial stressors. Mastery ranges from being unable to notice or acknowledge any psychological problems to utilizing specific strategies (e.g., behavioral activation, changing thoughts) to a more holistic response that integrates metacognitive knowledge to respond to psychosocial problems in personally acceptable ways.

Although beyond the scope of this paper, other models of metacognition have been explored within PTSD and trauma-related disorders ([70]). For example, the metacognitive model of PTSD has proposed that maladaptive beliefs about thoughts and traumatic memories following a traumatic event interrupt or impair natural emotion processing. This interruption is thought to exacerbate PTSD symptoms ([9]). Furthermore, metacognitive beliefs, meta-memory beliefs, and metacognitive control strategies have been suggested to play a role in maintaining PTSD symptoms.

### Metacognitive Changes in Trauma-Related Disorders

Metacognitive deficits have been observed in individuals who have experienced trauma. For example, when compared to a control group of individuals who have faced medical adversity, individuals with PTSD had significantly lower metacognitive mastery, or the ability to use metacognitive knowledge to respond to psychosocial challenges in increasingly complex and adaptive ways ([37]). Furthermore, lower mastery was linked to higher levels of distress and hyperarousal symptoms. In a sample of 51 individuals diagnosed with PTSD, it was also found that higher metacognition was related to lower levels of anger experience and expression of anger ([34]).

Importantly, metacognition has been found to play an important role in the experience of PTSD and related disorders. For example, one study found that for individuals with relatively higher levels of metacognition, the presence of positive world assumptions (e.g., the benevolence of people) was related to experiencing fewer PTSD symptoms. For individuals with lower metacognition, this protective relationship was not found ([1]). In another study, lower self-reflectivity was related to higher self-blame in individuals with PTSD ([15]). Finally, constructs closely related to metacognition, including mentalizing ([36]) and theory of mind ([59]), have been suggested as treatment targets in PTSD and trauma-related disorders.

## 5. Treatment Implications

We propose that narrative-oriented therapy for trauma-related disorders may lead to greater improvements in personal recovery and integration and that utilizing a metacognitively informed approach is especially important to foster these changes. [63] ([63]) recently outlined a set of tasks essential for the process of repairing one’s life narrative. This model focused on individuals who have been diagnosed with a wide range mental illnesses (e.g., psychosis, trauma-related disorders) and proposed that these individuals often have difficulty accepting the ill self, coping with the loss of the previous self, and imagining the future self. Thomsen et al. proposed four tasks for treatment and explored various obstacles to sharing one’s story within this framework. Tasks for treatment focused on narrative repair include (1) beginning to tell one’s life story, (2) narrating one’s personal experience with mental health challenges, (3) storying the past with a focus on sharing positive aspects of the self (e.g., the growing self), and (4) narrating the dreaming self. In this framework, Thomsen et al. describe common obstacles that may make sharing one’s life story difficult. These include (1) negative personal narratives or narratives that have an abundance of negative themes, (2) lack of coherence in one’s life narrative, and (3) the inability to synthesize and anchor narratives within the flow of one’s life and overarching narrative identity.

Of note, there is an existing literature that supports the use of narrative-based treatment for PTSD. For example, one recent meta-analysis of 35 studies found that narrative-based interventions are a viable and effective treatment option for individuals diagnosed with PTSD ([52]). Another study found that after repeated expressive writing about a traumatic event, trauma narratives tend to increase in coherence ([68]). As proposed earlier in this paper, a higher level of metacognition may be necessary for being able to utilize narrative intervention to promote change and recovery. For example, it has been suggested that individuals who construct narratives in self-reflective ways are better able to make meaning of traumatic experience in comparison to those who construct a narrative in ruminative ways (e.g., brooding, self-doubt) ([40]) or who draw meanings that lack a subjective perspective and personal detail ([64]). Overall, we suggest that metacognitive and narrative-focused therapy may offer a path forward for the treatment of chronic courses of PTSD that do not respond to existing evidence-based therapies or as a stepping-stone when preparing clients for beginning evidence-based trauma treatment that is focused on cognitive distortions, changing maladaptive thought patterns, or symptom improvement that may necessitate higher levels of metacognition.

### 5.1. Metacognitive Reflection and Insight Therapy

Metacognitive Reflection and Insight Therapy (MERIT) is one type of therapy that directly targets metacognition and incorporates reflection on life narratives. MERIT ([39]) is an individual psychotherapy that includes a set of eight core therapeutic elements that should be present in each session (see Table 2). These elements are divided into four content elements, two process elements, and two superordinate elements. The content elements include (1) a focus on the client’s agenda or seeking to understand the hopes, wishes, and plans a client brings to each session; (2) insertion of the therapist’s mind or openness to sharing one’s own thoughts and reactions to the client in a way that promotes reflective dialogue; (3) eliciting narrative episodes or focusing on the personal stories that make up the client’s life to understand past and immediate experiences; and (4) defining the psychological problem or outlining an agreed upon psychosocial challenge the client is experiencing. The two process elements include (5) reflection on the therapeutic relationship, which includes discussion of the interpersonal processes occurring within the session, and (6) reflection on client progress, which includes discussion of what is changing in the client’s life as a result of psychotherapy and if the client is achieving their goals. The final two elements include specific interventions related to metacognitive domains, including techniques to (7) stimulate self-reflectivity and understanding the mind of the other and (8) stimulate metacognitive mastery or the ability to use metacognitive knowledge to respond to psychosocial problems.

MERIT research has historically focused on individuals with serious mental illness (SMI) but is considered a cross-diagnostic treatment and has been used with a variety of individuals and clinical presentations. For example, one case study that focused on the overlap between schizophrenia and trauma found that not only did scores of metacognition increase after a course of MERIT, but this individual developed increasingly complex ideas about himself and others in his life, which he was able to use to respond to trauma-related challenges ([22]). Case studies have also focused on MERIT as a treatment for borderline personality disorder, which may have applications for the treatment of cPTSD. These case studies found that MERIT improves metacognition scores in individuals diagnosed with borderline personality disorder and ultimately assists these individuals with developing more complex understandings of self and using this knowledge to act as agents to respond to psychosocial challenges ([10]; [67]). In MERIT, clients may focus on a range of topics, including traumatic experiences and how these experiences have impacted their sense of self, their understanding of others, and their ability to engage in one’s community.

Although each therapeutic element is important in MERIT, the most relevant to the ideas expressed in this paper are elements related to eliciting narrative episodes and stimulating metacognitive domains of self-reflectivity, understanding the mind of the other, and mastery. For example, a focus on narrative episodes within MERIT is thought to combat fragmentation by allowing for additional opportunities to reflect upon and integrate specific experiences into the broader flow of life ([21]). As fragmentation decreases, a sense of narrative cohesion is likely to emerge, especially for previously fragmented or incoherent narratives of trauma. In addition, MERIT’s focus on eliciting narrative episodes allows for the application of narrative repair frameworks within an existing treatment ([63]). Complex relationships between personal narratives and each domain of metacognition also exist and have important implications for treatment implications.

#### 5.1.1. Self-Reflectivity

An important aspect of recovery from trauma is improving one’s understanding of and beliefs about the trauma narrative. Self-reflectivity includes a range of abilities that are relevant to this task, including the ability to identify discrete thoughts and emotions and to recognize that one’s beliefs may be inaccurate or incorrect. Without this capacity, it would be impossible to change one’s often complex and stable beliefs about traumatic experiences. Recovering from trauma can also require the acceptance of events or facts that cannot be changed, such as the loss of a friend in combat. If a person has a lower metacognitive ability, they may first need to reflect on discrete thoughts and emotions before they are able to recognize that thoughts or emotions have changed in response to the traumatic event. Higher levels of metacognition also allow a person to understand how traumatic events can impact their outlook on the self, others, and the world, and this may allow them to decide how to respond to their changed outlook. For example, someone who has experienced an interpersonal trauma might reflect on how this event affected their trust in others and has made it more difficult to build relationships. If the interpersonal trauma is ineffectively integrated (either not integrated or overly integrated), they may either neglect their needs for trust or focus too heavily on their lack of trust and give up on relationships. If they build their metacognitive ability, they may more effectively integrate this trauma into their sense of self (i.e., self-reflectivity) and use this knowledge to understand that they need relationships to move slowly and to communicate this to any new potential partners (i.e., mastery).

#### 5.1.2. Understanding the Mind of the Other

Having the ability to make judgments about others (i.e., understanding the mind of the other) can help people to re-evaluate trauma narratives and evolve their narratives to make guesses about others who may have participated in or contributed to the traumatic event. Improved ability to understand the mind of the other, including the intentions of a specific other’s nonverbal communication, may help individuals to better gauge the intentions of others as dangerous or benign and, by extension, help them to decide how to move forward in interpersonal relationships. For example, a veteran might have ended up in a dangerous situation because his commander ordered him to complete a mission. He may initially think the commander did not care about him or blame him, contributing to a reaction of anger. Through reflection with a therapist and building higher levels of metacognition, he might be able to recognize that the commander may have been under stress, may not have known the area was as dangerous as it was, or was ordered by someone else. This new perspective may help the veteran change his interpretation of the event, decrease his blame towards the commander, and improve his anger. Relatedly, improvements in decentration can help a person to acknowledge the potentially many interconnected factors that shape events, which is required in order to work through assumptions and attributions about traumatic experiences.

#### 5.1.3. Mastery

Finally, mastery speaks to the ability to cope with psychosocial challenges one is facing and plays a role in life narratives. Being able to sense, identify, and name a psychosocial challenge is an important step toward coping and foundational to key recovery processes, such as self-regulation and the development of agency. Stimulating mastery encourages individuals to move from generally passive coping strategies (e.g., avoidance of triggers) to more active strategies (e.g., engagement in treatment). As mastery increases, individuals may be better able to change their thoughts about a traumatic event or use specific metacognitive knowledge about themselves to cope and move forward with the flow of one’s life.

### 5.2. Using MERIT to Fill Treatment Gaps

In addition to functioning as a stand-alone treatment for PTSD, especially for those who have not responded to traditional, evidence-based PTSD treatment (e.g., cognitive processing therapy, prolonged exposure), MERIT can also serve as a “stepping-stone” intervention to prepare individuals for one of these therapies. One systematic review found that the pooled dropout rate from psychotherapy randomized controlled trials for PTSD was 16% ([35]), with dropout likely being much higher in real-world practice ([48]). Another study across three randomized controlled trials of CPT and PE reported a 30% dropout rate ([5]). Some predictors of dropout from trauma treatment include difficulty with emotion regulation and high levels of psychopathology ([7]). Another study showed that higher rates of avoidance predicted higher dropout rates from PTSD treatment ([61]). Some research found that describing more negative emotions experienced in the past and currently in the initial written narratives of CPT protected against dropout, while expression of highly generalized beliefs applied across time periods and situations predicted increased dropout risk ([2]). MERIT may help address predictors of dropout, specifically difficulties with emotion regulation and avoidance, to increase retention rates for “gold standard” PTSD treatment following an abbreviated course of MERIT.

For example, the metacognitive domain of self-reflectivity includes the ability to sense and identify a range of differently valanced emotions. In addition, metacognitive mastery is thought to improve the ability to experience emotion. In a study that reviewed 563 sessions of MERIT across 37 clients, it was found that emotional experience, expression, and regulation were associated with better outcomes, with emotional experience being associated with improvement in metacognitive mastery ([23]). Similarly, as discussed earlier, stimulating metacognitive mastery leads to improvement in the ability to cope with psychosocial stressors or challenges. For an individual who struggles with emotion regulation, stimulating mastery may take the form of learning specific coping skills to better manage emotions. In contrast, for an individual who tends to engage in avoidance, stimulating mastery may include taking steps towards using more active coping strategies.

MERIT may also represent a phase-based intervention to focus more holistically on connection and integration. The stage-based model of PTSD treatment proposes that there are three phases of PTSD treatment, including (1) safety and stabilization, (2) processing trauma, and (3) integration and connecting with others ([65]). Because MERIT focuses on integrating experiences and narratives into one’s evolving sense of self and better understanding specific others and one’s broader community, it may uniquely fill intervention gaps during phase three of PTSD treatment. 

## 6. Conclusions

Exposure to trauma can be a significant life event that can impact one’s personal life narrative and lead to clinical diagnoses, such as PTSD. In this paper, we reviewed alterations commonly found in trauma narratives. Although research is somewhat mixed, there is support for the idea that trauma narratives tend to be less cohesive and more fragmented than other life narratives. In addition, positive narrative themes (e.g., meaning-making attempts) are typically related to less severe symptomology. One framework that has been linked to narrative identity and may provide a way to better understand these alterations is the integrated model of metacognition. In this model, metacognition is defined as the ability to form increasingly complex and integrated ideas about oneself, others, and the world in addition to the ability to identity and manage psychosocial problems. Metacognitive deficits have been observed in individuals with trauma-related disorders; thus, metacognition may serve as an important intervention point to ameliorate these deficits and improve personal narratives. In fact, one therapy based on the integrated model of metacognition, Metacognitive Reflection and Insight Therapy (MERIT), has been shown to be an effective approach for a range of disorders, including trauma-related disorders. We propose that treatment focused on metacognition and narrative development may be especially beneficial for trauma-related disorders that have been unresponsive to other treatments, as a stepping-stone to help individuals become ready to engage in structured PTSD treatment, or as a method of increasing integration and connection. In this paper, we have outlined how experiencing trauma may impact personal narratives and explored metacognitive approaches that may improve a person’s understanding of trauma narratives, leading to increased integration and various other positive outcomes.

Although MERIT is an empirically supported treatment, limitations exist. For example, implementing MERIT requires extensive training and is not commonly utilized in a range of treatment settings at this time. This may lead to a lack of accessibility to the general population seeking MERIT-oriented care; however, the use of MERIT continues to expand and is becoming increasingly available. For example, group-based MERIT is being developed, which is likely to lead to increased availability and decreased cost of receiving a clinical dose of MERIT ([58]). Future directions for research include randomized control trials focused on the effectiveness of MERIT in PTSD and other trauma-related disorders, establishing a common metacognitive profile in trauma-exposed populations, and continuing to explore the impact of trauma narratives.

## Figures and Tables

**Table 1 behavsci-15-00150-t001:** Summary of alterations found in trauma narratives.

	Key Findings	Citation
Narrative Structure		
	Trauma narratives are less coherent than narratives of other key life events	([66]; [68])
	Trauma narratives are fragmented and incomplete immediately after a traumatic event and increase in coherence over time	([6]; [64])
	Overly integrated narratives (e.g., ineffective) are more highly associated with PTSD than narratives that lack integration	([56])
	Greater thematic coherence is associated with higher rates of anxiety	([66])
	The level of fragmentation in traumatic memories is equivalent to that of memories for non-traumatic events	([54])
	Trauma narrative coherence is related to trauma recovery	([26])
Narrative Content		
	Use of more emotion words, higher cognitive process, and less self-focus are related to lower trauma-related symptomology	([24])
	Framing meaning of trauma in terms of a personal, subjective response is protective against developing PTSD symptoms	([64])

**Table 2 behavsci-15-00150-t002:** Therapeutic elements of Metacognitive Insight and Reflection Therapy (MERIT).

	Element	Therapeutic Task
Content		
	Focus on the client’s agenda	Notice and reflect the client’s intentions and plans for each session (e.g., to feel safe)
	Insertion of the therapist’s mind	Share one’s thoughts and reactions to the client in a way that promotes reflective dialogue
	Elicit narrative episodes	Elicit and reflect on specific life narratives to increase coherence and identify themes
	Define the psychological problem	Define and explore a mutually agreed upon and understood psychosocial challenge
Process		
	Reflection on the therapeutic relationship	Discuss the interpersonal processes occurring within therapy (e.g., ruptures and repairs)
	Reflection on client progress	Discuss what is changing as a result of therapy and how the client has progressed
Superordinate		
	Stimulate self-reflectivity and understanding the mind of the other	Utilize metacognitive interventions to increase the client’s ability to reflect on and understand the self and others in complex ways
	Stimulate metacognitive mastery	Utilize metacognitive interventions to improve the client’s ability to manage and cope with psychological problems

## Data Availability

No new data were created or analyzed in this study. Data sharing is not applicable to this article.

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
