# Peer review of "Personal Narratives in Trauma-Related Disorders: Contributions from a Metacognitive Approach and Treatment Considerations"

_behavsci, 2025, doi:10.3390/bs15020150_

Round 1
Reviewer 1 Report
Comments and Suggestions for Authors
I invite the authors to extensively reconsider their paper.
Overall, the paper seems good, however, it lacks methodological rigour and specificity. The paper should be more structured following rigorous methodological recommendations. For better readability, the results should be summarized and presented in a form of a table. Please find specific comments below.
1. Despite the fact this is a narrative review, please make the paper more rigorous by following guidelines on conducting narrative reviews. For instance, these ones:
Sukhera J. (2022). Narrative Reviews: Flexible, Rigorous, and Practical. Journal of graduate medical education, 14(4), 414–417. https://doi.org/10.4300/JGME-D-22-00480.1
Pautasso M (2013) Ten Simple Rules for Writing a Literature Review. PLOS Computational Biology 9(7): e1003149. https://doi.org/10.1371/journal.pcbi.1003149
2. This paper lacks of summarized data presented in a form of a table. I feel the authors can create such tables, with (1) key points regarding Narrative Identity in Trauma-Related Disorders (with references), examples of studies, measures, and main outcomes; and (2) main points related to treatment and other recommendations. In the current form, the ideas presented are somewhat difficult to read. A more structured way of presenting results would be beneficial.
3. Your main aim was "This review aimed to expand upon and synthesize existing frameworks of narrative identity and metacognition", thus please indicate what are these frameworks with their specific characteristics, pros and cons. And as this was the main aim of the paper, please synthesize this in a form of a table.
In the current form, a mismatch between the aim and the paper narration exists.
4. Conclusions can be more specific. Please indicate not only what you have done, but also what exactly have you revealed in this review.
Some minor comments:
5. Minor: There is no need to use subsection 3.1 and 4.1., if you have no subsections 3.2 and 4.2.
6. Adding more keywords (up to 10) would be beneficial for better indexing.
7. Instead of using "persons", please use "people".
Please amend the paper comprehensively, particularly in terms of results presentation.
Author Response
I invite the authors to extensively reconsider their paper.
Overall, the paper seems good, however, it lacks methodological rigour and specificity. The paper should be more structured following rigorous methodological recommendations. For better readability, the results should be summarized and presented in a form of a table. Please find specific comments below.
Thank you for your thoughtful review of our manuscript. We have now structured our paper in a more rigorous way and clarified our aims, which we believe strengthens the paper and improves readability. Changes to the manuscript are highlighted in yellow.
- Despite the fact this is a narrative review, please make the paper more rigorous by following guidelines on conducting narrative reviews. For instance, these ones:
Sukhera J. (2022). Narrative Reviews: Flexible, Rigorous, and Practical. Journal of graduate medical education, 14(4), 414–417. https://doi.org/10.4300/JGME-D-22-00480.1
Pautasso M (2013) Ten Simple Rules for Writing a Literature Review. PLOS Computational Biology 9(7): e1003149. https://doi.org/10.1371/journal.pcbi.1003149
We reviewed both of these articles and found them to be helpful in improving our manuscript. Based on our reading of Sukhera, 2022 we made several changes to our manuscript, including 1) expanding and clarifying the aims and scope of the review, 2) better defining terms (e.g., PTSD, cognition), and 3) adding inclusion and exclusion criteria to our methods. Based on our reading of Pautasso, 2013, we aimed to 1) better define our topic of interest and intended audience and 2) improve readability of results. Please note that we now acknowledge that this review was intended to be theory driven (i.e., based within the integrated model of metacognition).
- This paper lacks of summarized data presented in a form of a table. I feel the authors can create such tables, with (1) key points regarding Narrative Identity in Trauma-Related Disorders (with references), examples of studies, measures, and main outcomes; and (2) main points related to treatment and other recommendations. In the current form, the ideas presented are somewhat difficult to read. A more structured way of presenting results would be beneficial.
We added a summary table of key findings related to alterations in personal narratives in individuals diagnosed with a trauma-related disorder. We believe this improves the readability of the manuscript. Because treatment implications are primarily related to a specific treatment (i.e., MERIT), we did not create a table related to treatment recommendations. We hope that the clarification of our aims supports and better explains this decision.
- Your main aim was "This review aimed to expand upon and synthesize existing frameworks of narrative identity and metacognition", thus please indicate what are these frameworks with their specific characteristics, pros and cons. And as this was the main aim of the paper, please synthesize this in a form of a table.
In the current form, a mismatch between the aim and the paper narration exists.
We significantly revised our aims and methods, which we believe better frames and improves the understandability of the paper. We hope that this clarifies our intentions for this paper and our decision to complete a theory-driven narrative review.
- Conclusions can be more specific. Please indicate not only what you have done, but also what exactly have you revealed in this review.
We expanded our conclusions to discuss what was revealed in this review.
Some minor comments:
- Minor: There is no need to use subsection 3.1 and 4.1., if you have no subsections 3.2 and 4.2.
We removed 3.1 and 4.1.
- Adding more keywords (up to 10) would be beneficial for better indexing.
We added additional key words, including post-traumatic stress disorder, psychotherapy, and MERIT.
- Instead of using "persons", please use "people".
Completed.
Please amend the paper comprehensively, particularly in terms of results presentation.
Reviewer 2 Report
Comments and Suggestions for Authors
This is a fairly well written report of an apparently selective literature review of an important and interesting topic, i.e., a narrative approach with a particular focus on meta-cognition as part of care for people with trauma related disorders (particularly posttraumatic stress disorder). Unfortunately, this paper's methodology is not rigorous enough as without a more systematic (scoping) review, unclear bias can mislead the reader (the authors' statement near the end of the paper that systematic reviews are needed does not explain why they did not conduct a scoping or systematic review).
Author Response
This is a fairly well written report of an apparently selective literature review of an important and interesting topic, i.e., a narrative approach with a particular focus on meta-cognition as part of care for people with trauma related disorders (particularly posttraumatic stress disorder). Unfortunately, this paper's methodology is not rigorous enough as without a more systematic (scoping) review, unclear bias can mislead the reader (the authors' statement near the end of the paper that systematic reviews are needed does not explain why they did not conduct a scoping or systematic review).
Thank you for taking the time to review our manuscript. Although we understand the importance of a rigorous methodology, we also value the importance of narrative reviews in exploring ideas and expanding theories. We have attempted to make our narrative review more rigorous by following recommendations from: Sukhera J. (2022). Narrative Reviews: Flexible, Rigorous, and Practical. Journal of graduate medical education, 14(4), 414–417. https://doi.org/10.4300/JGME-D-22-00480.1 and Pautasso M (2013) Ten Simple Rules for Writing a Literature Review. PLOS Computational Biology 9(7): e1003149. https://doi.org/10.1371/journal.pcbi.1003149. Specifically, we added clearer aims, inclusion and exclusion criteria, and better organized results to more clearly explore major findings based on this review. We hope that additional context and enhanced organization has improved this paper.
Reviewer 3 Report
Comments and Suggestions for Authors
he manuscript presents an engaging and well-articulated exploration of the topic; however, the following modifications may enhance its clarity and level of engagement.
- Please give the timeline for Ptsd to be termed chronic PTSD ( Line 74)
- Incorporating DSM-5 criteria could enhance non-medical readers' understanding of PTSD well.
- Treatment may also be more complicated in - Adverse childhood events( patients can also suffer from personality), female gender, and combat veterans ( who also might suffer Tbi )(Lines 101 to 110)
- Complex PTSD has been officially recognized in ICD-11, highlighting the need to understand its associated risk factors. There have been valuable case studies that shed light on this condition. I will provide a selection of references for further reading, and you can choose from these or select your own. 1 Ford JD, Courtois CA: Complex PTSD, affect dysregulation, and borderline personality disorder. Borderline Personal Disord Emot Dysregul. 2014, 1:9. 10.1186/2051-6673-1-9.https://doi.org/10.1002/jts.21699. Cutlip H A, Ang-Rabanes M, Mogallapu R (May 17, 2023) Unknown, Underserved, Underreported: A Case for Differentiation in Trauma Disorder Classification and Diagnosis. Cureus 15(5): e39157. doi:10.7759/cureus.39157.
- Typo error - 181 lines ( one months )
- The definition of cognition would be helpful for the readers.
- A visual representation of the pathophysiology of cognition in PTSD would be highly beneficial to understanding the condition more thoroughly. The right hemisphere is more predominant in emotional cognitions.
- The authors have provided a thorough explanation of MERIT, which is commendable. It would enhance the reader's understanding further if the authors could put this information into a table format below the explanation.
- If possible please also explain MERIT benifits in Boarderline personality and complex ptsd.
- In the conclusion section, it would be beneficial to explore the potential drawbacks of the therapy and its current accessibility for the general population. Additionally, we can discuss strategies to enhance its availability for high-risk groups, ensuring that more individuals can benefit from these therapeutic options.
Author Response
Reviewer #3
The manuscript presents an engaging and well-articulated exploration of the topic; however, the following modifications may enhance its clarity and level of engagement.
Thank you for your thoughtful review of our manuscript. We made numerous improvements based on your feedback and believe this has strengthened our paper. Changes to the manuscript are highlighted in yellow.
- Please give the timeline for Ptsd to be termed chronic PTSD (Line 74)
We now include information about the timeline for PTSD to be considered chronic.
- Incorporating DSM-5 criteria could enhance non-medical readers' understanding of PTSD well.
We agree that incorporating DSM-5 criteria for PTSD is helpful for readers. We now begin the paper with this information to better frame the rest of the manuscript.
- Treatment may also be more complicated in - Adverse childhood events( patients can also suffer from personality), female gender, and combat veterans ( who also might suffer Tbi )(Lines 101 to 110)
We added this information to the manuscript.
- Complex PTSD has been officially recognized in ICD-11, highlighting the need to understand its associated risk factors. There have been valuable case studies that shed light on this condition. I will provide a selection of references for further reading, and you can choose from these or select your own. 1 Ford JD, Courtois CA: Complex PTSD, affect dysregulation, and borderline personality disorder. Borderline Personal Disord Emot Dysregul. 2014, 1:9. 10.1186/2051-6673-1-9.https://doi.org/10.1002/jts.21699. Cutlip H A, Ang-Rabanes M, Mogallapu R (May 17, 2023) Unknown, Underserved, Underreported: A Case for Differentiation in Trauma Disorder Classification and Diagnosis. Cureus 15(5): e39157. doi:10.7759/cureus.39157.
These papers were very helpful in introducing and describing cPTSD, thank you for sharing. We added a review of each citation to text.
- Typo error - 181 lines ( one months )
Corrected.
- The definition of cognition would be helpful for the readers.
We added a broad definition of cognition and introduced the idea that metacognition falls under this umbrella.
- A visual representation of the pathophysiology of cognition in PTSD would be highly beneficial to understanding the condition more thoroughly. The right hemisphere is more predominant in emotional cognitions.
Thank you for the thoughtful suggestion to include a visual representation of the pathophysiology of cognition in PTSD. While we agree that such a figure could be beneficial for understanding the condition, we believe that this topic extends beyond the scope of our paper.
- The authors have provided a thorough explanation of MERIT, which is commendable. It would enhance the reader's understanding further if the authors could put this information into a table format below the explanation.
We added a table reviewing MERIT’s eight elements and hope that this enhances the reader’s understanding of MERIT in an easily digestible format.
- If possible please also explain MERIT benifits in Boarderline personality and complex ptsd.
We added a review of three relevant case studies to the MERIT section. Two focused explicitly on borderline personality disorder and one focused on an individual diagnosed with schizophrenia who had experiences significant trauma.
- In the conclusion section, it would be beneficial to explore the potential drawbacks of the therapy and its current accessibility for the general population. Additionally, we can discuss strategies to enhance its availability for high-risk groups, ensuring that more individuals can benefit from these therapeutic options.
We agree that this is an important point to consider. We expanded the conclusion to address this point and briefly explored the development of group-based MERIT as a potential way to expand accessibility.
Round 2
Reviewer 1 Report
Comments and Suggestions for Authors
Thank you for your comprehensive implementation of recommendations. The paper has been improved significantly.